# Optimization of Large Vessel Occlusion Detection in Acute Ischemic Stroke Using Machine Learning Methods

**DOI:** 10.3390/life12020230

**Published:** 2022-02-03

**Authors:** Gabor Tarkanyi, Akos Tenyi, Roland Hollos, Peter Janos Kalmar, Laszlo Szapary

**Affiliations:** 1Department of Neurology, Medical School, University of Pécs, 7624 Pécs, Hungary or tarkanyi.gabor@pte.hu (G.T.); or kalmar.peter@pte.hu (P.J.K.); 2Smart Data Group, E-Group ICT Software Zrt., 1027 Budapest, Hungary; akos.tenyi@egroup.hu (A.T.); roland.hollos@egroup.hu (R.H.)

**Keywords:** acute ischemic stroke, large-vessel occlusion, prehospital care, stroke scales, machine learning

## Abstract

The early detection of large-vessel occlusion (LVO) strokes is increasingly important as these patients are potential candidates for endovascular therapy, the availability of which is limited. Prehospital LVO detection scales mainly contain symptom variables only; however, recent studies revealed that other types of variables could be useful as well. Our aim was to comprehensively assess the predictive ability of several clinical variables for LVO prediction and to develop an optimal combination of them using machine learning tools. We have retrospectively analysed data from a prospectively collected multi-centre stroke registry. Data on 41 variables were collected and divided into four groups (baseline vital parameters/demographic data, medical history, laboratory values, and symptoms). Following the univariate analysis, the LASSO method was used for feature selection to select an optimal combination of variables, and various machine learning methods (random forest (RF), logistic regression (LR), elastic net method (ENM), and simple neural network (SNN)) were applied to optimize the performance of the model. A total of 526 patients were included. Several neurological symptoms were more common and more severe in the group of LVO patients. Atrial fibrillation (AF) was more common, and serum white blood cell (WBC) counts were higher in the LVO group, while systolic blood pressure (SBP) was lower among LVO patients. Using the LASSO method, nine variables were selected for modelling (six symptom variables, AF, chronic heart failure, and WBC count). When applying machine learning methods and 10-fold cross validation using the selected variables, all models proved to have an AUC between 0.736 (RF) and 0.775 (LR), similar to the performance of National Institutes of Health Stroke Scale (AUC: 0.790). Our study highlights that, although certain neurological symptoms have the best ability to predict an LVO, other variables (such as AF and CHF in medical history and white blood cell counts) should also be included in multivariate models to optimize their efficiency.

## 1. Introduction

Large-vessel occlusion (LVO) is present in 20–40% of acute ischemic stroke (AIS) cases, resulting in more severe symptoms and worse outcomes if not treated urgently [1]. In addition to well-established intravenous thrombolysis (IVT), experience using endovascular thrombectomy (EVT) to treat AIS patients with LVO is increasing [2]. However, the number of EVT-capable institutions, so-called comprehensive stroke centres (CSC), is limited [3]. The reliable detection of an LVO is currently only possible using radiological methods, primarily computed tomography angiography (CTA), which is mostly available in hospitals only [4].

Regarding patient pathways, two approaches have emerged. According to the first approach, AIS patients should first be transported to the nearest IVT-capable primary stroke center (PSC). If the presence of an LVO is confirmed, the patient is referred and transported to a CSC for EVT (drip-and-ship approach). In these cases, IVT could be started as soon as possible; however, the time spent in the PSC and the time of transportation may significantly delay the administration of EVT [5]. It should also be considered that IVT is only moderately effective if an LVO is present [6]. The second approach is to transport patients with a high likelihood of LVO directly to a CSC (mothership approach). This may slightly delay the start of the IVT due to the longer transportation time; however, it could significantly reduce the time to EVT administration [5].

One of the current major limitations of applying the mothership approach routinely is the deficit of easy-to-perform and sufficiently reliable prehospital methods to identify LVO [7]. Current stroke scales primarily focus on the assessment of clinical symptoms; however, other factors such atrial fibrillation (AF) in medical history or systolic blood pressure (SBP) may also have good predictive value [8,9]. The aim of our study was to comprehensively assess the associations between clinical symptoms, medical history variables, vital parameters, laboratory values and the presence of LVO in AIS, and to develop an optimal combination of them using machine learning tools and methods.

## 2. Methods

### 2.1. Study Cohort

A cross-sectional, observational study was performed based on a prospective registry of consecutive AIS patients presenting up to 4.5 h after symptom onset at the CSC of three university hospitals in Hungary (Appendix A) between November 2017 and July 2019. Data on medical history were collected from past medical documentation and based on personal interview with the patient and relatives upon arrival to the emergency department (ED) when possible. Baseline vital parameters and laboratory values were measured as a part of standard care. On admission, stroke symptoms and severity were assessed using the National Institutes of Health Stroke Scale (NIHSS). Detailed information on the registry is available in the Appendix A.

### 2.2. Outcome

Our outcome of interest was the presence of LVO on the on-admission CTA scan. Acute occlusions of the internal carotid artery (ICA), M1, M2 and M3 segments of the middle cerebral artery (MCA), A1 and A2 segments of the anterior cerebral artery (ACA), P1 and P2 segments of the posterior cerebral artery (PCA), basilar artery (BA), vertebral artery (VA) and tandem occlusions were considered according to Rennert et al. [10]. Scans were evaluated by trained neuroradiologist (who were blinded to clinical parameters) as a part of standard care. Patients who did not undergo CTA were excluded.

### 2.3. Statistical Analysis

Continuous variables were presented as mean and standard deviation (SD) or as median and interquartile range (IQR). Normality was assessed using the Shapiro–Wilk test and visually, based on Q–Q plots and histograms. Categorical variables were presented as counts and percentages. In the univariate analysis, a comparison of continuous variables was performed using a *t*-test or Mann–Whitney U test. Categorical data were compared using the Pearson X^2^ test or the Fischer exact test where appropriate. Receiver operating curve (ROC) analysis was used to assess the ability of variables and models to discriminate the presence of an LVO. The optimal cut-off score was calculated using the Youden J index.

### 2.4. Data Analysis

Data on 41 variables were collected and used for the modelling task. During pre-processing, variables were excluded from the analysis based on (i) having more than 20% missing values (Body temperature, SpO_2_), (ii) larger than 0.9 correlation with another variable (Hgb), and/or (iii) near zero variance (Extinction). Rows with missing values were omitted from the analysis. Variables were further processed with Yeo Johnson transformation to reduce skewness in lab variables and variables were centered and scaled to obtain statistical uniformity for machine learning (ML) modeling. Smote resampling was used to balance the sample difference in LVO and non-LVO groups. Grid search was used to select optimal hyperparameter for the models. For final model validation, a randomly selected hold-out test cohort was used consisting of 20% of the patient population. To assess the generalizability of the models a 10-fold cross validation was used. 

Four covariate groups were created based on the nature of variables including 6 baseline and demographic variables, 9 medical history variables with yes/no values, 10 laboratory variables with numeric values and 14 symptom-related variables with values on an ordinal scale. The predictive ability of these groups of variables was measured using binary logistic regression analysis and ROC analysis was performed based on probability values.

Feature selection was carried out using least absolute shrinkage and selection operator (LASSO) regression to determine the optimal combination of variables to predict LVO [11]. For further ML modeling, the selected variables were used only as covariates. The performance of three ML models—namely, logistic regression, random forest, and neural network—and elastic net method was compared with each other and with a logistic regression model with NIHSS as the only covariate using area under the ROC curve (AUC) statistic (see Appendix A). For neural network modeling, a multi-layer perceptron was used with one hidden layer of four neurons. Analysis was carried out in SPSS (version 26, IBM, New York, NY, USA) and R using the Caret ML library [12,13].

## 3. Results

A total of 646 patients were screened during the study period, 526 (81.4%) of whom underwent CTA imaging and were finally included in the analysis (46.2% female). The mean age of the study cohort was 68 ± 13 years; 227 patients had LVO (43.2%). The baseline characteristics of the study cohort and the ability of the variables to distinguish an LVO are presented in Table 1. NIHSS had the best discriminative ability with an AUC of 0.783 (95% CI: 0.742–0.824); the optimal cut-off value of NIHSS to detect an LVO was ≥9 points (sensitivity: 70.9%; specificity: 72.6%). The prevalence of several symptoms and the severity of symptoms were higher among LVO patients (Table 2.) The distribution of LVO location was as follows: 54 (23.8%) ICA, 74 (32.6%) MCA M1, 52 (22.9%) MCA M2, 4 (1.8%) MCA M3, 2 (0.9%) ACA, 1 (0.4%) PCA, 12 (5.3%) BA, 11 (4.8%) VA, and 17 (7.5%) tandem occlusions. The etiology of LVO strokes was more commonly cardioembolism and less commonly small-vessel disease, as compared to non-LVO cases (Table 1).

Regarding predefined covariate groups, the combination of symptoms had the best ability to discriminate an LVO (AUC: 0.779 on hold-out set and 0.785 after 10-fold cross validation; *p* < 0.001, respectively), followed by medical history (AUC: 0.602 and 0.686; *p* < 0.001), laboratory values (AUC: 0.637 and 0.641; *p* < 0.001) and baseline and demographic parameters (0.599 and 0.567; *p* < 0.001). NIHSS had an AUC of 0.783 and 0.790 after cross validation (*p* < 0.001).

### Data Driven Analysis

The results of the covariate group analysis showed that, over a combination of symptoms (NIHSS items), further variables could have potential discriminative power for LVO, especially among the anamnestic and laboratory related variables. Thus, we explored the potential of a mixed-covariate model for discriminate LVO patients using data-driven analysis and a variable selection process (see Appendix A). 

In the initial dataset, there was a relatively high amount of missing data (4% of the dataset), mainly at random properties (see Appendix A and related comments) and was mainly concentrated in a few variables. Our analysis showed that imputing missing values would negatively affect the performance of the final models (see Appendix A); thus, patients with missing values were omitted from the analysis and a two-step approach was followed to maximize sample size for modelling. After preprocessing the dataset, all samples with missing values were omitted (*n* = 293) and lasso regression was used to select the most predictive variables to LVO. Then, the final data-driven analysis was carried out using the original dataset, filtering only to these selected variables, and omitting patients with missing values (*n* = 483). During feature selection, a total of nine variables were selected for subsequent ML modelling (six symptom variables: language, facial palsy, LOC questions, visual field disturbance, gaze palsy and upper limb weakness; two medical history variables: atrial fibrillation (AF) and chronic heart failure (CHF); and one laboratory value: white blood cell (WBC) count).

Including the selected variables, four ML tools were applied: random forest (RF), logistic regression (LR), elastic net method (ENM), and simple neural network (SNN). The calculated AUC values on the hold-out set and after 10-fold cross-validation were 0.986 and 0.736 for the RF model, 0.816 and 0.775 for the LR, 0.813 and 0.773 for ENM and 0.808 and 0.772 for SNN.

## 4. Discussion

Our study has highlighted that the severity of certain neurological symptoms may have the best ability to predict an LVO, but our results also pointed out that other variables (notably, AF or CHF in medical history and on-admission WBC values) also have good predictive ability.

The clinical presentation of LVO in AIS is highly dependent on the site of occlusion [10]. Currently, NIHSS is the “gold-standard” for stroke severity assessment and has the best ability to detect LVOs—the previously reported AUC values were similar to our findings [7]. Despite the wide spectrum of symptoms assessed in NIHSS, it still occasionally fails to detect and assess posterior territory strokes appropriately. For short stroke scales, the challenge is to examine the full spectrum of symptoms corresponding to different vascular territory strokes without the process becoming too complicated. The results of a retrospective study suggested that cortical symptoms are better predictors of LVO than motor symptoms, but their combination has the highest accuracy [14]. Our findings showed that upper and lower extremity weakness had the best discriminative abilities, followed by gaze disturbance and facial palsy. However, it should be noted that the majority of the LVO cases in our study involved anterior circulation; therefore, the findings should be interpreted accordingly.

The use of ML methods to optimize prediction models is emerging in the field of stroke research to maximize the predictive performance of variable combinations [15]. Based on the previously mentioned findings, it is not surprising that feature selection using the LASSO method in our study mainly selected symptom variables (motor and cortical symptoms as well) for modeling. The selected symptoms represent a wide spectrum of LVOs in various vascular regions, as they mostly occur in anterior and posterior territory strokes as well. In addition, variables that had a strong association with the presence of LVO in the univariate analysis were selected—notably, AF, CHF, and WBC count. In a recent article by Wang et al. using a similar approach, a set of variables were initially selected based on research in the literature and clinical relevance for subsequent feature selection [15]. In contrast, in our study, we included all variables that were available in adequate quality from a multi-center registry. However, after feature selection, it appeared in both studies that, although symptoms provide the backbone of the models, other types of variables may be important factors and should be included as well.

Including these variables, all applied ML tools performed well on the full set of data (AUC > 0.800); however, after 10-fold cross validation, the performance of each markedly decreased and the AUC values of three models (RF, LR and ENM) ranged from 0.775 to 0.772; the SNN lagged slightly with an AUC of 0.736. The study by Wang et al. has applied a similar approach to optimize LVO prediction, and their results regarding the performance of ML tools were quite similar. The abilities of stroke scales for LVO detection has also been reported generally around this range in previous retro- and prospective studies [7,8,15].

Over recent years, a plethora of LVO detection methods have been developed and examined. For a tool to be applicable for prehospital use, several criteria must be met, such as high diagnostic accuracy, easy and fast application, user-friendliness, and cost-effectiveness [16]. The NIHSS may be too complex for routine prehospital use; therefore, the use of shorter scales is warranted at the cost of some reduction in accuracy. It should also be noted that some symptoms are not easily examinable by non-neurologists, such as gaze disturbance and visual field loss, two symptoms that were also selected for modelling in our study and, therefore, may limit prehospital applicability [17]. However, the inclusion of non-symptom variables is not common in LVO scales yet.

Regarding patient history and clinical parameters, a study has found that the history of AF and SBP ≤ 170 mm Hgmm are independent predictors of LVO in AIS, and these correlations were also confirmed by our results [9]. There have been some attempts to attach AF to various scales with heterogeneous results. A retrospective analysis has shown no improvement in the accuracy of four broadly used short stroke scales when AF was added as an element [18]. In contrast, another study found that the adding of AF to the Los Angeles Motor Scale (LAMS) could significantly improve its ability to detect LVOs [19]. In addition, several recently created LVO scales include AF as a variable [20,21]. The utility of including SBP in stroke scales is much less studied. A prospective observational study demonstrated that SBP may help to identify patients potentially eligible for EVT [22]. Chronic heart failure is an independent risk factor of stroke, and other diseases should be considered (such as AF, CAD and valvular disease) that are predisposing factors for CHF and AIS [23]. The association between CHF and the presence of LVO probably represents a wide spectrum of confounding and additive conditions. Therefore, CHF might be interchangeable or be combinable with the aforementioned cardiac diseases. Future studies may use a combined variable containing all predisposing cardiac diseases at once. 

Despite the amount of biomarker research in the field of AIS, so far, only a few markers that are potentially suitable for LVO detection have been identified. Our group has previously found an association between WBC counts and the presence of LVO which is also confirmed by the current investigation; however, the studied population was partially overlapping [24]. Other studies have revealed independent associations between protein markers (such as serum troponin and D-dimer) [25,26]. However, to date, they are not routinely used for screening in the prehospital setting.

Univariate analyses in our study revealed that the strength of associations between most variables and LVO is mild to moderate, the reason for which is probably that associations are affected by many known and unknown confounding factors (e.g., LVO location regarding symptoms). It is also clear that a combination of variables with such specificity cannot exceed a certain accuracy. The study highlighted that machine learning tools are extremely useful to reduce the dimensions of large datasets, and to assess and optimize predictive ability. However, the result should also be approached and interpreted from clinical and practical aspects as well, since the heterogeneity of clinical presentations may limit the clinical utility of these methods.

Molecular biomarkers supporting the clinical care of stroke, especially its classification and objective monitoring, are yet to be available. A better understanding of the biochemical and pathophysiological pathways and processes associated with LVO is needed to identify more specific biomarkers. Screening for a large number of potential biomarkers, i.e., the “omics” approach, and the combined analysis of multi-omic data, including proteomic, more recently glycomic, and metabolomic data, is a particularly promising solution for identifying new biomarkers. Extended stroke registers and multi-omic databases combining clinical and biomedical data are needed together with data analysis platforms that can facilitate to organize and analyze large amounts of data with modern machine learning methods, to identify new, complex biomarkers that support stroke typing and therapy monitoring [27,28].

It should also be noted that the definition of LVO is quite heterogenous, and previous studies and clinical trials have used various criteria for LVO classification [10]. Mechanical thrombectomy cannot be performed in some cases that are radiologically considered as cases of LVO. However, from a clinical aspect, the 2019 AHA/ASA stroke guidelines recommend considering MT in a wide spectrum of LVO cases. In the case of distal occlusions (e.g., MCA M2 and M3) and occlusions in the posterior circulation, the decision to indicate MT should be made on a case-by-case basis, weighing the potential costs and benefits [29]. Consequently, the scope of future studies should not only be the detection of LVO, but to detect the eligibility to MT early on.

Anterior and posterior circulation territory occlusions and strokes may show quite different clinical appearances and have different predisposing factors [30,31]. The NIHSS also investigates more anterior territory stroke symptoms and, thus, occasionally fails to correctly assess the severity of posterior strokes [32]. Although we aimed to create a universal LVO detection model in our study, we considered all types of LVO. However, for future studies, it may be worthwhile to optimize the prediction of anterior and posterior circulation LVOs separately in a similar way using ML methods, due to the aforementioned differences. Another possible direction is that, after performing a method optimized for anterior circulation LVOs, a method optimized for a posterior circulation LVOs should follow.

The main strength of our study is the comprehensive assessment of real-life, prospectively collected data from multiple centers using novel statistical methods that are not extensively used in medical research yet. However, our study also has some limitations. Firstly, the cross-sectional design only allows to assess associations but not causality. It is important to emphasize that potentially important variables may not have been included to the analyses due to multiple reasons (e.g., a large amount of missing data, or variables were not available in the stroke registry) which could have caused bias. In this study, we used 10-fold cross validation to estimate the generalizability and the true accuracy of the models; however, validation using an external dataset is needed to clinically validate our findings. Finally, ML tools function the best when applied to large datasets (“big-data”), which our dataset did not necessarily match.

## 5. Conclusions

The need for accurate LVO detection scales is emerging. A novel approach for this could be the machine-learning-based development of prediction models. Our study confirmed this, highlighting that neurological symptoms are the most useful to increase the accuracy of prediction models, but other types of variables (certain medical history data, and laboratory values) should also be included to maximize efficiency.

## Figures and Tables

**Table 1 life-12-00230-t001:** Baseline characteristics of the cohort according to the presence of LVO.

	LVO Present (*n* = 227)	LVO Absent (*n* = 299)	*p* Value	AUC (95% CI)
**Demographic characteristics**				
Age, years, median (IQR)	68 (61–79)	69 (59–77)	0.231	0.524 (0.467–0.582)
Gender, female, % (n)	49.8 (113)	43.5 (130)	0.151	0.530 (0.474–0.587)
**Elapsed times**				
Onset-to-ER assessment time, min, median (IQR)	83 (58–124)	88 (59–135)	0.110	-
ER assessment-to-CTA time, min, median (IQR)	14 (6–23)	17 (6–32)	0.043	-
**Parameters on admission**				
NIHSS score on admission, median (IQR)	12 (8–16)	6 (4–9)	<0.001	0.783 (0.742–0.824)
On admission SBP, mmHg, median (IQR)	160 (140–178)	169.5 (145–185)	0.005	0.420 (0.365–0.474)
On admission DBP, mmHg, median (IQR)	86 (78–99)	90 (80–100)	0.034	0.456 (0.401–0.511)
Heart rate, 1/min, median (IQR)	82 (72–93)	80 (71–92)	0.251	0.533 (0.477–0.589)
SpO_2_, %, median (IQR)	97 (96–98)	97 (96–99)	0.025	0.447 (0.345–0.550)
Body temperature, °C, median (IQR)	36.4 (36.0–36.5)	36.5 (36.2–36.6)	0.008	0.372 (0.270–0.474)
BMI, kg/m^2^, median (IQR)	25.78 (23.34–30.12)	26.72 (23.46–31.21)	0.125	0.447 (0.392–0.502)
**Laboratory parameters**				
Blood glucose, mmol/L, median (IQR)	6.90 (5.91–8.28)	6.50 (5.60–8.30)	0.084	0.548 (0.495–0.602)
INR, ratio, median (IQR)	1.03 (0.96–1.10)	1.00 (0.95–1.05)	<0.001	0.587 (0.534–0.640)
CRP, mg/L, median (IQR)	3.30 (1.50–7.20)	2.98 (1.55–5.80)	0.262	0.540 (0.486–0.595)
WBC, 10^9^/L, median (IQR)	8.62 (6.88–10.62)	7.94 (6.55–9.61)	0.005	0.583 (0.530–0.636)
Platelet, 10^9^/L, median (IQR)	233.5 (195–271)	224 (186–267)	0.078	0.532 (0.479–0.586)
Haematocrit, %, median (IQR)	40.0 (37.6–42.8)	41.1 (38.0–44.0)	0.034	0.449 (0.396–0.503)
Haemoglobin, g/dL, median (IQR)	138 (126–146)	141 (130–152)	0.005	0.433 (0.380–0.486)
Creatinine, µmol/L, median (IQR)	82 (69–99)	83 (69–101)	0.561	0.485 (0.431–0.539)
BUN, mmol/L, median (IQR)	6.26 (4.80–8.19)	6.10 (4.68–7.63)	0.173	0.527 (0.473–0.581)
AST, U/L, median (IQR)	20 (16–24)	20 (16–25)	0.480	0.476 (0.422–0.530)
ALT, U/L, median (IQR)	15 (11–22)	16 (12–22.5)	0.381	0.466 (0.412–0.520)
**Presence of vascular risk factors**				
Smoking, % (n)	34.9 (66)	31.4 (85)	0.424	0.517 (0.460–0.574)
Hypertension, % (n)	81.4 (180)	80.4 (234)	0.768	0.496 (0.439–0.553)
Diabetes mellitus, % (n)	21.5 (47)	28.6 (82)	0.069	0.475 (0.418–0.531)
Hyperlipidaemia, % (n)	59.2 (125)	58.3 (161)	0.840	0.495 (0.438–0.552)
Atrial fibrillation, % (n)	35.8 (78)	17.5 (50)	<0.001	0.590 (0.533–0.647)
Coronary artery disease, % (n)	29.6 (64)	21.9 (61)	0.051	0.535 (0.478–0.592)
Chronic heart failure, % (n)	17.9 (39)	8.9 (25)	0.002	0.549 (0.492–0.606)
Previous stroke/TIA, % (n)	21.0 (46)	23.2 (66)	0.564	0.494 (0.438–0.551)
Malignancy, % (n)	15.6 (33)	11.7 (33)	0.217	0.520 (0.462–0.577)
**Etiology (TOAST)**, % (n)			<0.001	
Large-artery atherosclerosis	26.4 (60)	27.8 (83)		
Cardioembolism	51.1 (116)	20.7 (62)		
Small vessel disease	0 (0)	21.7 (65)		
Other determined origin	0.4 (1)	5.0 (15)		
Undetermined etiology	22.0 (50)	24.7 (74)		

Abbreviation: LVO, large-vessel occlusion; AUC, area under the curve; CI, confidence interval; IQR, interquartile range; ER, emergency room; CTA, CT angiography; NIHSS, National Institutes of Health Stroke Scale; SBP, systolic blood pressure; DBP, diastolic blood pressure; BMI, body mass index; INR, International Normalized Ratio; CRP, C-reactive protein; WBC, white blood cell; BUN, blood urea nitrogen; AST, aspartate-aminotransferase; ALT, alanine-aminotransferase, TIA, transient ischemic attack.

**Table 2 life-12-00230-t002:** Distribution of symptom severity and prevalence as a function of LVO.

Symptoms (NIHSS Items)	Points			Presence			AUC (95% CI)
	LVO Present	LVO Absent	*p* Value	LVO Present	LVO Absent	*p* Value	
1A. Level of consciousness (LOC)	0 (0–0)	0 (0–0)	0.003	12.8%	5.4%	0.003	0.537 (0.487–0.587)
1B. LOC questions	1 (0–2)	0 (0–1)	<0.001	56.4%	33.1%	<0.001	0.638 (0.589–0.686)
1C. LOC commands	0 (0–2)	0 (0–0)	<0.001	47.1%	24.7%	<0.001	0.618 (0.569–0.667)
2. Gaze	0 (0–2)	0 (0–0)	<0.001	46.3%	15.1%	<0.001	0.666 (0.617–0.714)
3. Visual fields	0 (0–2)	0 (0–0)	<0.001	47.6%	21.4%	<0.001	0.632 (0.583–0.681)
4. Facial palsy	2 (1–2)	1 (0–2)	<0.001	85.9%	70.9%	<0.001	0.644 (0.597–0.692)
5. Arm weakness	3 (1–4)	1 (0–2)	<0.001	91.2%	72.6%	<0.001	0.738 (0.695–0.782)
6. Leg weakness	3 (1–3)	1 (0–2)	<0.001	83.3%	64.9%	<0.001	0.717 (0.671–0.762)
7. Limb ataxia	0 (0–0)	0 (0–0)	0.001	7.0%	17.4%	<0.001	0.450 (0.401–0.499)
8. Sensory deficit	0 (0–1)	0 (0–1)	0.688	26.9%	30.1%	0.418	0.492 (0.442–0.542)
9. Language/aphasia	1 (0–2)	0 (0–1)	<0.001	56.8%	37.1%	<0.001	0.634 (0.586–0.683)
10. Dysarthria	0 (0–1)	0 (0–1)	0.893	37.0%	38.1%	0.792	0.497 (0.447–0.547)
11. Extinction/inattention	0 (0–0)	0 (0–0)	0.001	9.7%	2.7%	0.001	0.535 (0.485–0.585)

Abbreviation: LVO, large-vessel occlusion; NIHSS, National Institutes of Health Stroke Scale; AUC, area under the curve; CI, confidence interval.

## Data Availability

The data presented in this study are available on request from the corresponding author. The data are not publicly available due to patient privacy considerations (HIPPA).

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
