# Peer review of "Optimization of Large Vessel Occlusion Detection in Acute Ischemic Stroke Using Machine Learning Methods"

_life, 2022, doi:10.3390/life12020230_

Round 1

Reviewer 1 Report

A very well-prepared original article. Needs minor revisions on English grammar and words.

Line 45: resulting ------ resulting in

Line 47: patient------ patients

Line 79: On admission strokes---- On admission, strokes

Line 100: preprocessing------- pre-processing

Line 114: these group------- these groups

Line 229: Heterogenous-------- heterogeneous

Line 257: Metobolomic----- metabolomic

Author Response

Dear Reviewer,

Thank you for your honorable feedback on our manuscript.

We have fixed all the mentioned errors in the text.

Reviewer 2 Report

This well-written manuscript provides clinically relevant information on an important topic.

Some comments are below:

1. In Result section: LVO occlusion location,  4 (1.8%) MCA M3 is considered LVO; as we know M3 occlusion is not LVO. Will this change the results of the model? Because the outcome measure is incorrect?

2. Please Kindly explain the rationale for considering M2 occlusion as LVO? Not every M2 occlusion patient would benefit from ECR, especially for the M2 occlusion, but with a minor deficit patient(low NIHSS score). (Dobrocky et al., 2021)

3. The NIHSS is heavily weighted towards hemispheric stroke (anterior circulation stroke); the NIHSS does not adequately assess symptoms for Posterior circulation stroke. (Schneck, 2018; Sommer et al., 2018). This study combines the posterior and anterior circulation stroke. Have the authors considered that the NIHSS score (not an ideal measurement for posterior circulation stroke) is less sensitive for the posterior circulation stroke? Please explain the rational of combining the posterior and anterior circulation stroke.

4.Did the author validate the model with an external dataset, and what is the accuracy of the predicting model?

5.Please kindly explain the clinical application? We know that the patients with server NIHSS/ clinical symptoms are more likely to have LVO. Can we tell from the model what is the Cut of the NIHSS to predict an LVO? For example, if a patient with NIHSS 8, how likely would the patient have an LVO?

Author Response

Dear Reviewer,

Thank you for your time and effort to review our manuscript.

Hereby we provide a point-by-point response to the comments.

Points 1 and 2:

Without a doubt it is not easy to correctly define what exatly an LVO is. Radiological definition of LVO is not neccesarily match clinical relevance (eg. suitability for mechanical thrombectomy (MT)).

Our approach was to include every type of LVO that is potentially suitable for mechanical thrombectomy (MT) (even if in selected cases only). The decision of including M2 and M3 occlusions was based on the 2019 AHA/ASA guidelines (Powers et al, Stroke 2019) (recommenadtion 3.7.2. / 3) that recommends to consider MT for selected patients with M2 or M3 occlusions (COR: IIb).

To highlight this point we have added a section to the Discussion (Lines 265 - 273).

Point 3:

Thank you for highlighting this point. It is true that NIHSS may have less power to detect posterior territory strokes. We also mention this in the manuscript (Lines 186-188).

However our consideration was the same in this case as described in the previous point: to include every type of LVO that might be suitable for MT. The 2019 AHA/ASA stroke guidelines reccomend to consider MT in patients with vertebral, basilar and posterior cerebral artery occlsuions (reccomendation 3.7.2. / 5; ROC: IIb).

One of the aims of our study was to see if including other variables in stroke scales and using machine learning methods can help to improve the detection of such occlusions, therefore resulting a more broad utility.

Point 4:

For validation we have used no external validation, but 10-fold cross-validation (which has almost the same accuracy in our case we believe).

Point 5:

Our main aim was to find out how machine learning methods can optimize big datasets to maximize their utility for LVO detection. We were also curios to see if we could outperform the "gold-standard" NIHSS with this data-based approach.

We have to main findings: firstly that machine learning (ML) methods included non-symptom variables in the models, its clinical utility can be that stroke scales should include non-symptoms variables as well in the future.

The other main finding is ML based models were not able to outperform the NIHSS. The reason of it is discussed in the Discussion section. The main clinical utility of this finding is highlighting the necessity to research more specifict LVO markers.

The disadvantage of ML methods is that due to their complexity we cannot exatly assign scores to patients, rather just calculated probabilities of LVO. Therefore only AUC values are presented and exact cut-off scores were not calculated.

In the case of NIHSS we have calculated the optimal cut-off point and related sensitvity and specificty values, and the results were added to the Results section.

Reviewer 3 Report

modeling (6 symptom variables: language, facial palsy, LOC questions, visual field disturbance, gaze palsy and upper limb weakness; 2 medical history variables: atrial fibrillation (AF) and chronic heart failure (CHF) and 1 laboratory value: white blood cell (WBC) count) in lines 168-171. 9 variables to predict the large vessel occlusion. AF related to infection with high WBC.

The symptoms of LVO were severer, 6 variables seem to be many .

Is LVO absent small vessel disease, embolic stroke or undifferential type?

LVO mixed the anterior and posterior circulation, it seems to be confusing.

The logistic regression can do forward selection to find the good predictors. This study lacks external validation and needs add on.

The AUC of NIHSS score with 0.783 (0.742-0.824) was better than your model.

The AUC of coronary artery disease with 0.535 (0.478-0.592) is similar with the AUC of chronic heart failure with 0.549 (0.492-0.606), but it was not included in the model. The vascular events related to large vessel occlusion.

Author Response

Dear Reviewer,

Thank you for your time and effort to comment our manuscript.

Hereby we present a detailed step-by-step response to the comments.

Point 1:

Thank you for highlighting this point. Now we have included data (in Table 1.) on etiologies in the two group of patients (LVO vs. non LVO), and have included an interpretation in the Results section (Lines 137-138).

Point 2:

Reviewer 2 has also mentioned this aspect.

The reason for including posterior circulation LVOs as well was that we wanted to include every patient who is potentially eligible for mechanical thrombectomy (MT). The latest AHA/ASA guidelines (Powers et al. in Stroke, 2019) state that MT should be considered in patients with occlusions in the posterior circulation (reccomendation 3.7.2. / 5; level IIB).

We believe that an optimal LVO detection method should also be able to detect posterior LVOs. We included a brief section on this in the Discussion (Lines 266-274).

Point 3:

Forward selection logistic regression is a good method to reduce dimensions of models, however it lacks further optimization of the selected variables. In this study we used LASSO regression, which also features variable optimization (e.g. transformations where adequate.

For validation of the models we have not used an external dataset, but 10-fold cross validation (using randomly picked subsets of data for validation multiple times and calculating the true value of predictive ability from those results).

Point 4:

Our hypothesis was that using machine learning (ML) methods we can reduce the number of variables that should be included in LVO prediction models, and at the same time we can create models that outperform NIHSS.

Our findings had highlighted that we can reduce the number of variables, but cannot overperform NIHSS, however our result show that ML based models seems to be non-inferior to NIHSS.

The main clinical utility of these results is that future stroke scale studies should also include non-symptom variables. Besides it seems that more LVO specific markers are needed to be discovered.

Point 5:

Coronary artery disease was removed from the model by the LASSO regression, probably due to high level of multicolienarity and simultenaous occurence with chronic heart failure.

Round 2

Reviewer 2 Report

  1. In the method section, the author mentioned the outcome of interest was the presence of LVO on the admission CTA : “Our outcome of interest was the presence of LVO on the on-admission CTA scan. 85 Acute occlusions of the internal carotid artery (ICA), M1 and M2 segments of the middle cerebral artery (MCA), A1 and A2 segments of the anterior cerebral artery (ACA), P1 and P2 segments of the posterior cerebral artery (PCA), basilar artery (BA), vertebral artery (VA) and tandem occlusions were considered according to Rennert et al [10]” did not include M3. But in the results section the LVO included M3 : “Distribution of LVO location 136 was the following: 54 (23.8%) ICA, 74 (32.6%) MCA M1, 52 (22.9%) MCA M2, 4 (1.8%) 137 MCA M3, 2 (0.9%) ACA, 1 (0.4%) PCA, 12 (5.3%) BA, 11 (4.8%) VA, and 17 (7.5%) tandem 138 occlusions.)”? Please kindly explain.

  1. The authors added the stroke classification (TOAST) in Table 1; there are 3 LVO patients were classified as small vessel disease. I am curious about the occlusion location for these small vessel disease patients who also were LVOs. As I know, it is unlikely that the patients were small vessel disease and also with Large vessel occlusion. 

Author Response

Dear Reviewer!

Thank you for your comments, hereby we would like to present the answers.

Comment 1:

Thank you for highlighting this point. We made a mistake here. MCA M3 occlusions were icluded in the analysis, however, somehow it was forgotten to mention this in the Methods section.

By now we have corrected the Methods section.

Comment 2:

Thank you for mentioning this point. We have checked this cases, all of this patients had M2 occlusions and were from the same stroke centre. We have consulted the corresponding physicians from that centre. According to them the reason for categorizing the etiology as small-vessel disease in these cases was the deficit of cortical symptoms, the presence of a lacunar size infarction and the deficit of large-artery atherosclerosis or cardiogenic risk factors of embolism. However, the fact that these patients had an LVO were not taken into account.

These cases were re-evaluated by two independent experts and the final conlusion was that the etiology of these LVO cases were re-categorized as undetermined origin strokes, based on the fact that the presence of LVO makes small-vessel origin very unlikely.

Reviewer 3 Report

Unfortunately, the revision of the manuscript seems to be not satisfying yet.

  1. The etiology showed not the same population with cardioembolism predominant and lack of p to explain the same or different population.
  2. The posterior circulation with severe symptoms with higher NHISS, needs to be removed in this study.
  3. It is a severe bias without external validation.
  4. The age and NHISS would predict ischemic stroke well, but this article was not better than basic prediction.
  5. The author replied that coronary artery disease was removed from the model by the LASSO regression, probably due to high level of multicolienarity and simultenaous occurence with chronic heart failure. But it is out of clinical practice for atherosclerosis, it must remove chronic heart failure rather than coronary artery disease.

Author Response

Dear Reviewer,

Thank you for reviewing our manuscript, we endeavor to respond appropriately and satifactorily to all comments. Please find the responses below.

Comment 1:

Thank you for noticing this mistake. We have calculated the p using the Pearson Chi square test and now included the result in Table I. The results showed significant difference in the distribution of etiologies regarding the two groups.

Comment 2:

We agree that there may be relevant differences in the appearance of anterior and posterior territory LVO-s. However we would like to emphasize that the aim of our research was to see iff we can develop a method that can detect both anterior and posterior territory LVO-s as well (thus it is an "universal" LVO detection method).

We consulted with the authors and there was a consensus that it would be worthwhile to examine the cases of anterior and posterior area LVOs separately using a similar method. However, we have come to the conclusion that this would be beyond the scope and of the present work and would exceed the quantiative limits.

However, due to the importance of this issue we have included a new paragraph in the Discussion in which we discuss this topic and the highlighted concerns, and the potential directions future research (Line 284-292).

Comment 3:

We agree that external validation is the "gold-standard" of validating the utility of previously well-defined diagnostic methods. However, the main scope of our study was not to clinically validate a new method, rather to present a new way of using machine learning methods to reduce dimensions of LVO detection tools.

We did 10-fold cross-validation to "validate" our findings, however in our case the aim of this was to find the estimated "true accuracy" of the models. We did not wanted to describe a model that is already finalized for clinical use (this will also be mentioned in the response for comment 5).

We would like to emphasize this more in the manuscript as well, therefore we supplemented the Limitations section. (Line 299-302)

Comment 4:

Our results showed slightly smaller (but not significantly different) AUC than NIHSS, however the aim of our study was to provide a simplified model that provide similar performance to this general stroke scale and work as a potential alternative. NIHSS is not suitable for routine pre-hospital use as it may be too complex and time-consuming.

For this reason, our conclusion was not that we were able to develop a better method than NIHSS, but rather that it may be necessary to include non-symptomatic variables in the short LVO detection scales.

Comment 5:

We agree that certain variables may be more relevant (for example the presence coronary artery disease  (CAD) may also indicate carotis atherosclerosis). However, we followed previously described statistical methhods that brought out chronic heart failure (CHF), which we respected.

We have tried to change CHF to CAD in the final model: the obtained AUC value in the new model was minimally lower (not significantly). Therefore we can say that these two variables can be interchangeble.

We have wrote a new paragraph about this in the Discussion. (Lines 241-247).